# LRR Receptor-like Protein in Rapeseed Confers Resistance to *Sclerotinia sclerotiorum* Infection via a Conserved *Ss*NEP2 Peptide

**DOI:** 10.3390/ijms26104569

**Published:** 2025-05-10

**Authors:** Chenghuizi Yang, Weiping Zhong, Wei Li, Yunong Xia, Lei Qin, Xianyu Tang, Shitou Xia

**Affiliations:** 1Department of Agriculture and Forestry, Hainan Tropical Ocean University, Sanya 572022, China; ychz121@hntou.edu.cn; 2Hunan Provincial Key Laboratory of Phytohormones and Growth Development, College of Bioscience and Biotechnology, Hunan Agricultural University, Changsha 410128, China; zweiping@stu.hunau.edu.cn (W.Z.); weili@cqnu.edu.cn (W.L.); yun0623@stu.hunau.edu.cn (Y.X.); leiqin@stu.hunau.edu.cn (L.Q.); chuxuantingnasha@stu.hunau.edu.cn (X.T.); 3College of Life Science, Chongqing Normal University, Chongqing 401331, China

**Keywords:** *Brassica napus*, LRR-RLP, plant immunity, *Sclerotinia sclerotiorum*, *Ss*NEP2, *Bna*SOBIR1

## Abstract

*Brassica napus* is one of the most extensively cultivated oilseed crops in China, but its yield is significantly impacted by stem rot caused by *Sclerotinia sclerotiorum*. Receptor-like proteins (RLPs) and receptor-like kinases (RLKs) play essential roles in plant–pathogen interactions; however, their regulatory mechanisms remain largely unknown in *B. napus*. In this study, we investigated the function of the leucine-rich repeat receptor-like protein *Bna*RLP-G13-1 in *Brassica napus* immunity. Previous observations indicated that *B. napus* plants expressing *Bna*RLP-G13-1 exhibited enhanced resistance to *Sclerotinia sclerotiorum*. We hypothesized that *Bna*RLP-G13-1 mediates pathogen recognition and immune signaling. To test this, we employed mitogen-activated protein kinase (MAPK) activity assays, transgenic overexpression analyses, and pathogen infection assays. Our results demonstrated that *Bna*RLP-G13-1 recognizes the conserved necrosis- and ethylene-inducing peptide *Ss*nlp24*_SsNEP2_* derived from *S. sclerotiorum*, triggering MAPK cascades and subsequent immune responses. Furthermore, protein interaction studies revealed that *Bna*RLP-G13-1 physically interacts with the receptor-like kinase *Bna*SOBIR1, which is essential for full antifungal defense activation. These results elucidate the molecular basis of *BnaRLP-G13-1*-mediated immunity, providing insights into improving disease resistance in oilseed crops.

## 1. Introduction

Over the course of evolution, plants have developed a sophisticated surveillance system to monitor pathogen invasion and stimulate their own immunity [1]. Studies have demonstrated that higher plants contain numerous immune receptors, distributed on the cell surface and within cells, which detect various pathogenic molecules associated with infections [2,3]. Many pattern recognition receptors (PRRs) on the surface of plant cells recognize signal molecules with unique conserved motifs, triggering the first layer of the innate immune response, which is called pathogen-associated molecular pattern (PAMP)-triggered immunity (PTI). PRRs consist of two types of cell surface proteins: receptor-like proteins (RLPs) and receptor-like kinases (RLKs), both of which contain leucine-rich repeats (LRRs) [4].

To date, an increasing number of RLPs have been found to play roles in disease resistance in *Arabidopsis thaliana*. For example, *Atrlp1* did not recognize the enigmatic microbe-associated molecular pattern (MAMP) of Xanthomonas (eMAX) and stimulated the PTI immune response, indicating that *At*RLP1 is the specific recognition receptor of eMAX and participates in the process of plant immunity [5,6]. *At*RLP32 recognized bacterial translation initiation factor 1 (IF1), induced the plant immune response, and enhanced resistance to *Pseudomonas syringae* pv. *tomato (Pst)* strain DC3000 in *A. thaliana* [7]. Similarly, *At*RLP23 has been reported to recognize conserved sequences in necrosis- and ethylene-inducing peptide 1 (Nep1)-like proteins (NLPs), which are widely present in pathogens, thereby activating immune responses. The overexpression of *AtRLP23* in potato enhanced plant resistance to *Sclerotinia sclerotiorum* [8]. Spraying small peptides synthesized from conserved sequences of NLPs in *Botrytis cinerea* improved the resistance of *A. thaliana* to *Alternaria brassicicola* [9]. However, not all nlp24 (a conserved 24-amino-acid fragment found in most NLPs) small peptides can be recognized by *At*RLP23. For example, the conserved nlp24 small peptide in *Pythium oligandrum* stimulated plant resistance to pathogens, but not through the *At*RLP23 pathway [10,11]. A conserved small peptide in *S. sclerotiorum* necrosis- and ethylene-inducible peptide 2 (*Ss*NEP2) increased plant resistance to biotrophic oomycetes, but its mechanism remains unknown [12].

Due to the lack of a kinase domain, RLPs can only recognize pathogens, and their signal transduction requires the formation of complexes with specific LRR-RLKs [13,14]. Arabidopsis LRR-RLK SOBIR1 (suppressor of brassinosteroid-insensitive 1-associated receptor kinase 1 (BAK1)-interacting receptor-like kinase 1) and its homologs form complexes with different RLPs to transmit signals. For instance, *At*SOBIR1 forms a complex with *At*RLP23 and *At*BAK1 to activate the NLP peptide immune response [8], and *At*RLP30 forms a complex with *At*BAK1 to activate the immune response to sclerotinia culture filtrate elicitor1 (SCFE1) in the fungal cell wall [15]. In addition, the signaling processes of *At*RLP1, *At*RLP32, and *At*RLP42 require SOBIR1 [5,16]. The tomato receptors *Verticillium* effector 1 (Ve1), the *Cladosporium fulvum* resistance protein that is produced by secreting the effector avirulence protein 4 (Cf-4), and the *Lycopersicon esculentum* ethylene-inducing xylanase 1 (*Le*Eix1) and *Le*Eix2 also interact with the SOBIR1 homolog in tomato in response to fungal infection [17].

*Brassica napus* is an important oilseed crop planted globally for vegetable oilseed production, animal feed, and biofuel [18,19]. Sclerotinia stem rot (SSR), caused by *S. sclerotiorum*, is a devastating disease in oilseed rape that seriously affects yield and quality [20,21]. In China, yield losses in oilseed rape caused by SSR typically range from 10% to 20%, but can reach up to 80% during severe outbreak seasons [22]. Despite its devastating impact, the molecular mechanism behind the interaction between *B. napus* and *S. sclerotiorum* is still largely unknown. Oilseed rape is one of the main hosts, and the *Bna*RLP family may play a crucial role in resistance to *S. sclerotiorum* infection. Therefore, we conducted a series of studies to analyze the regulatory role of the leucine-rich repeat receptor-like protein *BnaRLP-G13-1* in the interaction between *B. napus* and *S. sclerotiorum*. Our results indicate that *BnaRLP-G13-1*, in combination with *Bna*SOBIR1, enhances the resistance of *B. napus* to *S. sclerotiorum* by recognizing a conserved amino acid sequence derived from *Ss*NEP2.

## 2. Results

### 2.1. BnaC04g56380D Overexpression Enhances Plant Resistance to S. sclerotiorum

In our previous study, we observed that *BnaC04g56380D* expression was upregulated in both resistant and susceptible oilseed rape cultivars following *S. sclerotiorum* infection [23]. Consistent with this, in the moderately resistant cultivar (Xiangyou 15, XY15), the expression of *BnaC04g56380D* was significantly increased after 48 and 96 h of *S. sclerotiorum* infection, indicating that *BnaC04g56380D* was induced after infection and might positively regulate resistance to *S. sclerotiorum* in *B. napus* (Figure 1A). To improve the disease resistance of XY15, we selected it as the target material for the cloning and resequencing of *BnaC04g56380D*, a candidate gene potentially involved in *S. sclerotiorum* resistance. Similarly to *AtRLP23*, *BnaC04g56380D* contained only one exon without an intron (Appendix A). Although *BnaC04g56380D* had a small number of base variations, which caused individual amino acid changes compared with the database sequence, its secondary structure domain was not changed and still belonged to the RLP family (Appendix A). Subcellular localization analysis indicates that *Bna*C04g56380D is localized to the plasma membrane (Figure 1B).

We then further overexpressed *BnaC04g56380D* in *A.* thaliana and B. *napus* to explore its function. While the overexpression of *BnaC04g56380D* did not affect the growth and development of *A. thaliana* (Appendix A), the lesion area of the 35S- *BnaC04g56380D*/Col overexpression plants was much smaller than that of Col-0 after *S. sclerotiorum* infection (Figure 1C,D). Furthermore, when *BnaC04g56380D* was overexpressed in XY15, the growth phenotype of the identified T1 generation of the overexpression lines was not affected (Appendix A). According to the RT-qPCR results, the lines with the highest overexpression of *BnaC04g56380D* were selected for *S. sclerotiorum* infection testing. As shown in Figure 1E, the lesion areas of the plants overexpressing *BnaC04g56380D* were significantly smaller, at only about 50% the size of those of the wild-type XY15 plants (Figure 1F). These results demonstrate that the overexpression of *BnaC04g56380D* can effectively improve plant resistance to *S. sclerotiorum*.

### 2.2. BnaC04g56380D Shares Functional Similarity with AtRLP23

A phylogenetic tree constructed using the maximum likelihood method revealed that *Bna*C04g56380D clustered closely with *At*RLP23, suggesting a high degree of homology between the two proteins [23]. However, further validation was required to determine whether *BnaC04g56380D* shared the same function as *AtRLP23*. Mutants of *AtRLP23* were generated using CRISPR gene-editing technology or obtained as T-DNA insertion mutants (Appendix A). Infection assays revealed that the lesion areas in the *rlp23-1* and *rlp23-3* mutants were larger than those in Col-0, indicating an increased susceptibility to *S. sclerotiorum* (Figure 2A,B). When *BnaC04g56380D* was introduced into *rlp23-1*, the immunodeficiency caused by the *AtRLP23* mutation was restored (Figure 2D,E). Mitogen-activated protein kinase (MAPK) cascades serve as an important signaling pathway in plant immune responses and play a critical role in *Brassica napus* defense against *Sclerotinia sclerotiorum* infection [24]. To determine whether MAPK cascades were activated in *rlp23-1* and *rlp23-3*, Western blot analysis was performed. As shown in Figure 2C, the MAPK pathway was induced normally in the complemented plants containing *BnaC04g56380D*, demonstrating that *BnaC04g56380D* functions similarly to *AtRLP23*. Since *Bna*C04g56380D and *At*RLP23 belong to the same group (Group 13) in the phylogenetic tree [23], we designated *BnaC04g56380D* as *BnaRLP-G13-1* (*BnaRLP-Group13-1*).

### 2.3. BnaRLP-G13-1 Enhances Plant Immunity by Recognizing Ssnlp24_SsNEP2_

In a previous study, we found that the 24 conserved amino acids (*Ss*nlp24*_SsNEP2_*) from *Ss*NEP2 could enhance plant resistance to *Hyaloperonospora arabidopsidis* Noco2 in *A. thaliana* [12], but the specific mechanism for this remained unknown. To further explore whether the enhancement of plant immunity by *Ss*nlp24*_SsNEP2_* was related to *At*RLP23, the expression levels of *AtRLP23* and related genes after small-peptide treatment were detected. After *Ss*nlp24*_SsNEP2_* treatment for 4 h, the expression level of *AtRLP23* was found to be increased, and it continued rising over time (Figure 3A). The transcript level of *AtSOBIR1*, which formed a complex with *At*RLP23 to transmit signals, was significantly upregulated after small-peptide treatment (Figure 3A). Another essential component of the signaling pathway, *AtBAK1*, was also enhanced slightly (Figure 3A). These results illustrate that *Ss*nlp24*_SsNEP2_* can increase the expression level of *AtRLP23* and its related genes.

Then, the interaction between *At*RLP23 and *Ss*nlp24*_SsNEP2_* was detected by bimolecular fluorescence complementation (BiFC). As shown in Figure 3B, a fluorescence signal appeared in the cells of the leaves co-transformed with *Ss*nlp24*_SsNEP2_*-YCE and YNE-*At*RLP23, indicating that there was a direct interaction between *Ss*nlp24*_SsNEP2_* and *At*RLP23 and that the interaction occurred on the plasma membrane. Similarly, a direct interaction between *Ss*nlp24*_SsNEP2_* and *Bna*RLP-G13-1 also occurred on the plasma membrane. However, no direct interaction was found between *Ss*NEP2 and *At*RLP23.

To further check whether *At*RLP23 is the sole receptor of *Ss*nlp24*_SsNEP2_* in *A. thaliana*, *Ss*nlp24*_SsNEP2_* was evenly sprayed on the leaf surface of the *AtRLP23* deletion mutants. The results showed that *Ss*nlp24*_SsNEP2_* could cause a strong MAPK cascade reaction in Col-0 (Figure 4A). Nevertheless, *Ss*nlp24*_SsNEP2_* still slightly activated MAPK cascades in the *rlp23-1* and *rlp23-3* mutants, although these activations were far less than that in Col-0. The above results indicate that *Ss*nlp24*_SsNEP2_* may interact with other receptors in plants to activate MAPK cascades, in addition to *At*RLP23. After treatment with *Ss*nlp24*_SsNEP2_*, the lesion area on the surface of Col-0 was reduced, suggesting that *Ss*nlp24*_SsNEP2_* could enhance resistance in Arabidopsis to *S. sclerotiorum* (Figure 4B,C). However, the infection results showed no significant difference in lesion areas on the leaf surface of the *rlp23-1* or *rlp23-3* mutants compared to the control, regardless of *Ss*nlp24*_SsNEP2_* treatment (Figure 4B,C). This evidence indicates that *Ss*nlp24*_SsNEP2_* mainly interacts with *At*RLP23 to induce plant immunity and that *Ss*nlp24*_SsNEP2_* cannot enhance plant resistance to *S. sclerotiorum* after *AtRLP23* deletion. Additionally, the results of the ROS burst assay indicate that, in the absence of RLP23, *Ss*nlp24*_SsNEP2_* failed to induce a reactive oxygen burst in plants (Figure 4D).

### 2.4. Identification of BnaRLP-G13-1 Downstream Protein BnaSOBIR1

*At*SOBIR1, as the binding protein of *At*RLP23, forms a complex with *At*BAK1 to trigger the immune response. Since *BnaRLP-G13-1* shares functional similarity with *AtRLP23*, we hypothesized that it might interact with SOBIR1 to mediate immune signaling. To test this, we used the *At*SOBIR1 protein sequence as a query to perform BLASTP searches against the *B. napus* genome. Based on sequence similarity and domain composition, we identified six potential *Bna*SOBIR1 homologs, which are summarized in Table 1. The evolutionary relationship among these candidate genes was then analyzed, and their encoded proteins were found to have formed two branches during the evolution process. For better distinction, the branches including *Bna*A03g14760D and *Bna*C03g17800D were collectively referred to as class I and named “*Bna*SOBIR1 Group A”; the other branch was collectively referred to as class II and named “*Bna*SOBIR1 Group B” (Figure 5A). The results of the protein sequence alignment showed that the kinase domains of these six candidates were highly conserved, with differences observed only before the transmembrane domain (Appendix A), indicating that these candidate genes may have a protein kinase activity similar to that of *AtSOBIR1*. Two genes were then randomly selected from the two branches for functional verification. *BnaA03g14760D* was selected to represent *Bna*SOBIR1 Group A, while *BnaA04g18590D* represented *Bna*SOBIR1 Group B. After the cloning and resequencing of *BnaA03g14760D* and *BnaA04g18590D* from XY15, we found that some base substitutions occurred in *BnaA03g14760D* and *BnaA04g18590D*, but these substitutions did not affect their secondary structure formation compared with the database sequences (Supplementary Appendix A).

*AtSOBIR1* was originally discovered during the screening of *AtBIR1* inhibitors. As *AtBIR1* negatively regulates plant immunity, and its deletion mutants have obvious disease-resistant phenotypes, the *sobir1bir1-1* double mutant suppresses the disease resistance phenotype caused by *AtBIR1* deletion to a certain extent [25]. Based on this, the *sobir1bir1-1* double mutant was transformed with the *BnaSOBIR1* gene. As shown in Figure 5B, either *BnaA03g14760D* or *BnaA04g18590D* could restore the double mutant phenotype to a *bir1-1* single mutant. The transformants became dwarf again, with curled and senescent leaves. The Trypan blue staining results showed, in the 35s-*BnaSOBIR1*/*sobir1bir1-1* plants, the reappearance of a high level of cell death (Figure 5C). Furthermore, the *PR* genes of 35s-*BnaA03g14760D*/*sobir1bir1-1* and 35s-*BnaA04g18590D*/*sobir1bir1-1* were significantly upregulated, to the same level as the *bir1-1* single mutant (Figure 5D,E). Together, these results prove that *BnaA03g14760D* and *BnaA04g18590D* also play similar roles to the homologous gene of *AtSOBIR1*. According to the results of the phylogenetic tree (Figure 5A), *BnaA03g14760D* and *BnaA04g18590D* were subsequently named as *BnaSOBIR1-1* and *BnaSOBIR1-2*, respectively.

### 2.5. BnaSOBIR1 Improves Plant Resistance to S. sclerotiorum

The expression level of *BnaSOBIR1 Group A* did not change significantly after 48 h of *S. sclerotiorum* infection. However, 96 h after inoculation, its expression was significantly upregulated, reaching approximately three times the pre-inoculation level. On the contrary, the overall expression of *BnaSOBIR1 Group B* was downregulated after *S. sclerotiorum* inoculation (Figure 6A). To explore whether *BnaSOBIR1-1* and *BnaSOBIR1-2* were both involved in the plant immune response, they were overexpressed in Arabidopsis. As expected, the overexpression of *BnaSOBIR1-1* and *BnaSOBIR1-2* did not affect the growth of the transformants (Appendix A). Moreover, it was found that the overexpression of either *BnaSOBIR1-1* or *BnaSOBIR1-2* resulted in reduced lesion areas in the plants, which were approximately 25% of those observed in the Col-0 plants (Figure 6B,C). This indicates that both *BnaSOBIR1-1* and *BnaSOBIR1-2* can enhance plant resistance to *S. sclerotiorum*.

Similarly, when *BnaSOBIR1-1* and *BnaSOBIR1-2* were overexpressed in XY15, it was observed that their overexpression did not impact the normal growth and development of *B. napus* (Appendix A). Subsequently, the lines with the highest levels of *BnaSOBIR1-1* and *BnaSOBIR1-2* overexpression in XY15 were selected for a pathogen infection assay. As shown in Figure 6D,E, the lesions on the leaf surfaces of 35S-*BnaSOBIR1-1*/XY15 and 35S-*BnaSOBIR1-2*/XY15 were significantly smaller than those on XY15, indicating that the overexpression of *BnaSOBIR1-1* or *BnaSOBIR1-2* in oilseed rape could improve plant resistance to *S. sclerotiorum*. Thus, it can be concluded that both *Bna*SOBIR1-1 and *Bna*SOBIR1-2 play a role in the interaction between plants and *S. sclerotiorum*, significantly enhancing plant resistance to this pathogen.

### 2.6. BnaRLP-G13-1 Interacts with BnaSOBIR1 on Plasma Membrane

To determine whether *Bna*RLP-G13-1 physically interacts with *Bna*SOBIR1, as previously reported for *At*RLP23 and *At*SOBIR1 in *A. thaliana* [8], BiFC assays were conducted. As expected, fluorescence was observed in the leaf cells at the transformation site following the co-transformation of YNE-*BnaRLP-G13-1* and YCE-*BnaSOBIR1* in tobacco (Figure 7A). This confirms that *Bna*RLP-G13-1 can interact with both *Bna*SOBIR1-1 and *Bna*SOBIR1-2, with the interaction occurring at the plasma membrane.

To further verify protein interactions occurring on the plasma membrane of the cell, a yeast two-hybrid membrane system (Y2H) was selected and Y2H experiments were conducted. As shown in Figure 7B, yeast co-transformed with pPR3N-*Bna*SOBIR1-1 and the target vector could grow normally on the SD/-Leu-Trp-His-Ade media, and yeast co-transformed with the bait vector containing the target gene and pPR3N-*Bna*SOBIR1-2 could also grow normally (Figure 7B). Together, these results illustrate that *Bna*SOBIR1-1 and *Bna*SOBIR1-2 can interact with *Bna*RLP-G13-1.

## 3. Discussion

Our previous findings indicated that *BnaC04g56380D*, a member of the *BnaRLP* family, was upregulated upon *S. sclerotiorum* infection in *B. napus* [23]. This observation has drawn our attention to its potential role in plant defense. To optimize germplasm resources and enhance disease resistance in rapeseed, we selected XY15 (a high-quality, moderately resistant cultivar) as a sample for the further investigation of *BnaC04g56380D*. Following *S. sclerotiorum* infection, the expression level of *BnaC04g56380D* was significantly upregulated in XY15, suggesting its positive regulatory role in the interaction between *B. napus* and *S. sclerotiorum*. Resequencing results revealed that *BnaC04g56380D* was transcribed as an intronless gene, a feature it shares with *AtRLP23*, which is also composed of a single exon. Secondary structure predictions confirmed that *BnaC04g56380D* in XY15 retained the essential structural domains of the RLP family and was localized to the plasma membrane. The heterologous overexpression of *AtRLP23* in potato can increase the resistance of *S. sclerotiorum* [8]. Similarly, overexpression of *BnaC04g56380D* in both *A. thaliana* and *B. napus* enhanced plant resistance to *S. sclerotiorum*.

Given that *BnaC04g56380D* and *AtRLP23* belonged to the same phylogenetic group, we hypothesized that they might share similar functions. To verify this, we performed a genetic complementation assay using the *rlp23-1* mutant. Theoretically, the lesion size in the complemented lines should have been similar to that of Col-0; however, the transgenic *35S-BnaC04g56380D/rlp23-1* plants exhibited significantly smaller lesion areas than the wild type (WT), which might be attributed to the overexpression of *BnaC04g56380D* in the *rlp23-1* background. RLP23 can specifically recognize nlp24 to activate MAPK cascades [8]. Western blot analysis revealed that *BnaC04g56380D* was able to activate the MAPK signaling cascade and restored the recognition of the nlp24 peptide, a function that was lost in the *rlp23-1* mutant. Notably, the transgenic lines exhibited excessive MAPK activation, which we speculated to be due to the overexpression of *BnaC04g56380D*. It is important to note that MAPK activation was evaluated relative to the transgenic plants themselves rather than *Col-0*.

Although other *AtRLP* members were also present within the same phylogenetic group, here, we focused solely on the role of *BnaC04g56380D* in disease resistance and named *BnaC04g56380D* as *BnaRLP-G13-1*. Whether *BnaC04g56380D* exhibits functional similarity with other *AtRLP* members remains to be investigated.

As reported in our previous study [12], *Ss*NEP1 is not secreted under normal conditions and does not participate in the infection process of *S. sclerotiorum*, whereas *Ss*NEP2 is secreted and causes visible damage to plant leaves. Therefore, we selected its conserved sequence for further investigation. Albert et al. found that both the full nlp24 sequence and a shorter 20-amino-acid fragment could trigger plant immune responses [8]. However, as the 24-amino-acid sequence is a conserved motif in NLPs and is widely used in research, this study employed nlp24 and *Ss*nlp24*_SsNEP2_* peptides to explore their functions. Consistently, *Ss*nlp24*_SsNEP2_* could induce the expression of *AtRLP23*, *AtSOBIR1*, and *AtBAK1*, although the expression levels varied over time. *AtRLP23* may function as a receptor for the recognition of *Ss*nlp24*_SsNEP2_*, thereby eliciting a sustained upregulation in its expression. *At*SOBIR1 acted as a co-receptor binding to *At*RLP23, and its expression level was affected by *At*RLP23. In addition, *AtSOBIR1* played an important regulatory role in plant immunity as a suppressor of *AtBIR1* and a co-receptor of multiple RLPs [5,16]. Therefore, its expression is strictly regulated, and a sustained high expression level might cause cell death [16]. Compared with *At*SOBIR1, brassinosteroid-insensitive 1-associated receptor kinase 1 (BAK1) had more binding proteins and was also involved in the regulation of brassinosteroid [26]. The prolonged high expression of BAK1 may lead to increased cell death. Despite differences in their expression levels, *AtRLP23*, *AtSOBIR1*, and *AtBAK1* were all upregulated at the onset of *Ss*nlp24*_SsNEP2_* induction. This suggests that *Ss*nlp24*_SsNEP2_* might activate plant immunity through the RLP23-SOBIR1-BAK1 signaling pathway.

The BiFC results showed that *Ss*nlp24*_SsNEP2_* could interact with *At*RLP23 in Arabidopsis; however, *Ss*NEP2 could not be directly recognized by *At*RLP23. A previous study showed that *Pya*NLPs could bind GIPC to exert its cytolytic activity [27], and *Ss*NEP2, as a member of the NLP family, might exert its toxic effect to promote *S. sclerotiorum* infection in a similar way. In this scenario, it is plausible to speculate that NLPs secreted by pathogens might be cleaved by enzymes evolved from host plants during the interaction processes between plants and *S. sclerotiorum*, exposing the conserved 24-amino-acid sites and triggering plant immunity after being recognized by RLP23, thereby enhancing resistance to *S. sclerotiorum*.

As MAPK cascades could still be activated by *Ss*nlp24*_SsNEP2_* spraying on *rlp23-1* or *rlp23-3*, we suggest that there may be another receptor that can recognize *Ss*nlp24*_SsNEP2_* in Arabidopsis. In plants, MAPK cascades not only participate in the immune process but also play an important role in responding to abiotic stress [28]. However, the lesion areas on the leaf surfaces of the mutants treated with *Ss*nlp24*_SsNEP2_* were not significantly different from those on the WT, indicating that *Ss*nlp24*_SsNEP2_* could not enhance plant resistance to *S. sclerotiorum* without *AtRLP23*. The ROS burst results also indicate that *Ss*nlp24*_SsNEP2_* cannot induce a reactive oxygen burst in plants without *AtRLP23*. Based on these results, we suggest that *Ss*nlp24*_SsNEP2_* mainly interacts with *At*RLP23 to trigger plant immunity. Although *Ss*nlp24*_SsNEP2_* might be recognized by other receptors to activate MAPK cascades, it still cannot effectively activate the plant immune response.

*At*RLP23 needs to combine with *At*SOBIR1 to form a complex so as to transmit the recognition signal of a pathogen downstream and activate the plant immune response [8]. The orthologs of *AtSOBIR1* in *B. napus* naturally formed two branches during evolution, which might be due to sequence differences before the transmembrane domain, and the protein kinase domain after the transmembrane domain showed high degrees of similarity and identity, indicative of protein kinase activity conservation in both branches. As a suppressor of BIR1, SOBIR1 can inhibit the function of BIR1 in plants [25]. BIR1 is a negative regulator of plant immunity, and its mutant *bir1-1* exhibits obvious disease-resistance-related phenotypes [25]. Compared to *bir1-1*, the growth phenotype of the double mutant *sobir1bir1-1* more closely resembles that of the wild type with reduced disease resistance [25]. After transforming *BnaSOBIR1-1*/*BnaSOBIR1-2* into *sobir1bir1-1* double mutants, it was found that the phenotypes of *bir1-1* reappeared for the complementation lines, indicating that both were homolog genes of *AtSOBIR1* in *B. napus*. Although only two genes were selected as representatives of the two branches of *Bna*SOBIR1 Group A and *Bna*SOBIR1 Group B for testing, it is reasonable to speculate that the remaining *BnaSOBIR1* genes may have the same important function because of their high sequence similarity and identity.

The overexpression of *BnaSOBIR1-1* and *BnaSOBIR1-2* in *A. thaliana* and *B. napus* improved resistance to *S. sclerotiorum*, and their effects were more significant than those in the *BnaRLP23* overexpression plants. This may be because SOBIR1 is a binding protein of various RLPs, so its overexpression can improve not only the RLP23-mediated pathway, but also other RLP pathways involved in the regulation of plant immunity, and the synergistic nature of multiple pathways produces cumulative effects. Similarly, oilseed rape *Lep*R3 can recognize AvrLm1 (*Leptosphaeria maculans* effector 1) in the blackleg pathogen and interact with SOBIR1 to trigger the plant immune response [29]. Both Y2H and BiFC assays verified that *Bna*SOBIR1-1 and *Bna*SOBIR1-2 could interact with *Bna*RLP-G13-1. Together, these experimental results prove that in *B. napus*, *Bna*SOBIR1 also acts as a co-receptor to bind to *Bna*RLP-G13-1, thereby activating the downstream immune response and enhancing the plant’s resistance to pathogens.

## 4. Materials and Methods

### 4.1. Identification and Sequence Analysis of BnaRLP-G13-1 and BnaSOBIR1

*AtRLP23* (*AT2G32680*)/*AtSOBIR1* (*AT2G31880*) genomic sequence and annotation data were downloaded from TAIR (http://www.arabidopsis.org, accessed on 1 January 2022). *BnaRLP-G13-1*/*BnaSOBIR1* genomic sequence and annotation data were downloaded from Ensembl plants (http://http://plants.ensembl.org/index.html, accessed on 1 January 2022). The multiple sequence alignment of *Bna*SOBIR1 was performed using Clustal x1.83 software and opened with MEGA7.0 software; then, the maximum likelihood (ML) method was used to construct an evolutionary tree. Bootstrap was set to 1000. The domains contained in the sequence were analyzed by PredictProtein (https://predictprotein.org/, accessed on 1 June 2022).

### 4.2. Fungal Strains and Plant Materials

The wild-type strain 1980 was subcultured on potato dextrose agar (PDA) in an incubator at 20 °C for daily storage and 4 °C for long-term storage. Because the *AtRLP23* T-DNA insertion mutant (salk_034225) ordered from (Arashare) was named as *rlp23-1* [30], the *AtRLP23* CRISPR mutant generated in this study was named as *rlp23-3*. Knockout mutants, overexpression plants, and complementation plants were transformed with the vector into Col-0 (or *rlp23-1*) through Agrobacterium-mediated transformation [31]. The overexpression lines of *B. napus* were generated via tissue culture using Agrobacterium-mediated hypocotyl transformation [32]. The obtained plants were subjected to antibiotic screening and PCR identification. The *A. thaliana* and *B. napus* (XY15 or its transformant, Xiangyou 15, a high-quality variety of *Brassica napus* planted widely in the middle and lower reaches of the Yangtze River in China, referred to as XY15 in this article) used for the *S. sclerotiorum* virulence assays were cultured in a growth room at 22 °C with 16 h of light and 8 h of darkness; 4-week-old (*A. thaliana*) or 150-day-old (*B. napus*) plants were used for the assays.

### 4.3. Pathogen Infection Assay

Mycelial plugs (2 or 5 mm in diameter) from *S. sclerotiorum* (wild-type) were harvested from the edges of colonies maintained in PDA cultures for 48 h and then inoculated on leaves, with 2 mm plugs for *A. thaliana* and 5 mm plugs for *B. napus*. Lesion areas were measured 48 h later and counted with ImageJ. Two leaves were used per experiment, and each experiment was repeated three times. The data were analyzed using SPSS Statistics v.24.0 (IBM, Armonk, NY, USA). Statistical significance was determined using Student’s *t*-test, with *p*-values < 0.05 considered significant.

To analyze *Ss*nlp24*_SsNEP2_*-induced pattern-triggered immunity, leaves from 4-week-old Arabidopsis were sprayed with 1 μM *Ss*nlp24*_SsNEP2_* (the small peptide was synthetically produced and used after being diluted with ddH_2_O). After 24 h, seedlings were inoculated with *S. sclerotiorum* (wild-type). Lesion areas were measured 36 h later and counted with ImageJ.

### 4.4. MAPK Activity Assay

Two-week-old Arabidopsis seedlings grown on 1/2 MS medium plates were sprayed with 1 µM *Ss*nlp24*_SsNEP2_*, and samples were collected 0 and 15 min after each treatment. Then, protein extract buffer was added to extract the total protein. The phosphorylation of MAPK was detected via Western blot analysis with anti-pERK (no. 4370S, 1:2500 dilution; Cell Signaling Technology, Boston, MA, USA). Ponceau staining served as the internal reference. All experiments were repeated at least 3 times.

### 4.5. RT-qPCR Analysis

The gene expression levels of *BnaC04g56380D*, *BnaSOBIR1* Group A, and *BnaSOBIR1* Group B were analyzed at 48 and 96 h after inoculation with *S. sclerotiorum* (wild-type). Leaf samples (*B. napus*, XY15) were collected from a 5 mm area surrounding the inoculation site, including necrotic tissue. In contrast, the expression levels of *AtRLP23*, *AtSOBIR1*, and *AtBAK1* were examined at 4 and 24 h following *Ss*nlp24*_SsNEP2_* treatment, with samples collected from the entire leaf (*A. thaliana*, Col-0). The RNA of the test samples was extracted using an Eastep^TM^ Super Total RNA Extraction Kit, and the GoScript™ Reverse Transcription System Kit was used for cDNA synthesis (Promega, Madison, WI, USA). Quantitative expression assays were performed using the SYBR^®^ Green Premix Pro Taq HS qPCR Kit II (AG11702, Accurate Biotechnology (Hunan) Co., Ltd., Changsha, China) with the StepOneTM Real-time PCR Instrument Thermal Cycling Block. The primers were designed by the Primer Premier 5.0 program and amplicon-sized between 100 and 400 bp (Appendix A). The internal references were *Actin1* for *A. thaliana* and *UBC9* for *B. napus* [33].

### 4.6. Reactive Oxygen Species Burst Assay

The ROS burst assay was conducted following the previously described procedure, with slight modifications [34]. Leaf disks from true leaves from five-week-old plants (*A. thaliana*, Col-0) grown in soil were kept in the dark overnight with 200 µL of sterile distilled water in a 96-well plate. A reaction solution (170 μL) including 100 µM of luminol, 20 µg/mL of horseradish peroxidase, and 1 µM of synthetic peptides (nlp24 and *Ss*nlp24*_SsNEP2_*) was added into each well just before measurement. Luminescence was recorded at 2-minute intervals using Spark®(Tecan, Männedorf, Swiss Confederation) over a duration of approximately 60 minutes. Each treatment involved the measurement of six biological replicates, and the experiments were independently repeated three times.

### 4.7. Bimolecular Fluorescence Complementation (BiFC)

For the BiFC assays, the full-length cDNA of *BnaRLP-G13-1* was amplified via PCR using specific primer pairs (Appendix A), inserted into the plasmid pSPYNE digested with *Kpn*I and *Bam*HI, and *BnaSOBIR1* was cloned into a pSPYCE plasmid digested with *Kpn*I and *Bam*HI. pSPYNE and pSPYCE represent two segments of the yellow fluorescent protein (YFP), with YNE containing the N-terminus of the YFP and YCE containing the C-terminus of the YFP. The constructs for bimolecular fluorescence complementation (BIFC) were transformed into *Agrobacterium tumefaciens* GV3101 via electroporation. *A. tumefaciens* cultures carrying recombinant plasmids were diluted with infiltration buffer (10 mM MgCl_2_, 10 mM MES, 150 μM AS, pH = 5.6) to obtain OD600 = 1. Then, equal amounts of each culture were mixed and used for infiltration. Leaves (*N. benthamiana*) were imaged for BiFC after 36–72 h infiltration (Axio Imager 2, ZEISS, Oberkochen, Germany). YFP was excited at 470–490 nm and acquired at 500–540 nm.

### 4.8. Yeast Two-Hybrid (Y2H) Assays

For the yeast two-hybrid (Y2H) assays, the full-length cDNAs of *BnaRLP-G13-1* were amplified via PCR using specific primer pairs (Appendix A) and then inserted into the plasmid pBT3-STE digested with *Sfi*I to generate bait constructs. *BnaSOBIR1* was cloned into a pPR3-N plasmid digested with *Sfi*I to create the prey construct. All the prey and bait constructs were co-transformed into competent cells of the NMY51 yeast strain. Yeast transformants expressing each pair of proteins were assayed for growth on SD/-Leu/-Trp/-His/-Ade. pNubG-Fe65 + pTSU2-APP was used as the positive control, and pGADT7-T and pGBKT7-Lam served as the negative controls.

### 4.9. Trypan Blue Staining

Two-week-old Arabidopsis seedlings grown on 1/2 MS medium plates were soaked in 1% Trypan blue solution for 1 h and then decolorized with bleaching solution and photographed (Stemi 508, ZEISS, Oberkochen, Germany).

### 4.10. Subcellular Localization Assay

To assess the subcellular localization of *BnaC04g56380D* in *N. benthamiana*, a *BnaC04g56380D*-eGFP fusion gene driven by the 35S promoter was employed. The construct was introduced via *Agrobacterium tumefaciens* strain GV3101-mediated transformation, and the bacterial suspension was infiltrated into the abaxial epidermal cells of four-week-old leaves [35]. Fluorescence microscopy was performed 48–72 h post-infiltration using an Axio Imager 2 fluorescence microscope (ZEISS). GFP was excited at 470–490 nm and acquired at 500–540 nm.

## 5. Conclusions

In this study, we found that the overexpression of *BnaC04g56380D* (*BnaRLP-G13-1*) enhanced plant resistance to *S. sclerotiorum* and further investigated its underlying mechanism. *Bna*RLP-G13-1 contributed to plant immunity by recognizing a conserved amino acid sequence in *Ss*NEP2. Additionally, we screened for downstream proteins that interacted with *Bna*RLP-G13-1 in *B. napus* and analyzed their roles during infection. The overexpression of *BnaSOBIR1-1* and *BnaSOBIR1-2* in both *A. thaliana* and *B. napus* improved resistance to *S. sclerotiorum*, and these proteins were found to interact with *Bna*RLP-G13-1. Based on these findings, we propose that *Bna*RLP-G13-1, as a membrane surface receptor, recognizes *Ss*nlp24*_SsNEP2_* and forms a complex with *Bna*SOBIR1 and *Bna*BAK1, transmitting signals to activate downstream immune responses (Figure 8). As a key immune signal transducer, *Bna*RLP-G13-1 plays a crucial regulatory role in the interaction between rapeseed and *S. sclerotiorum*.

## Figures and Tables

**Figure 1 ijms-26-04569-f001:**
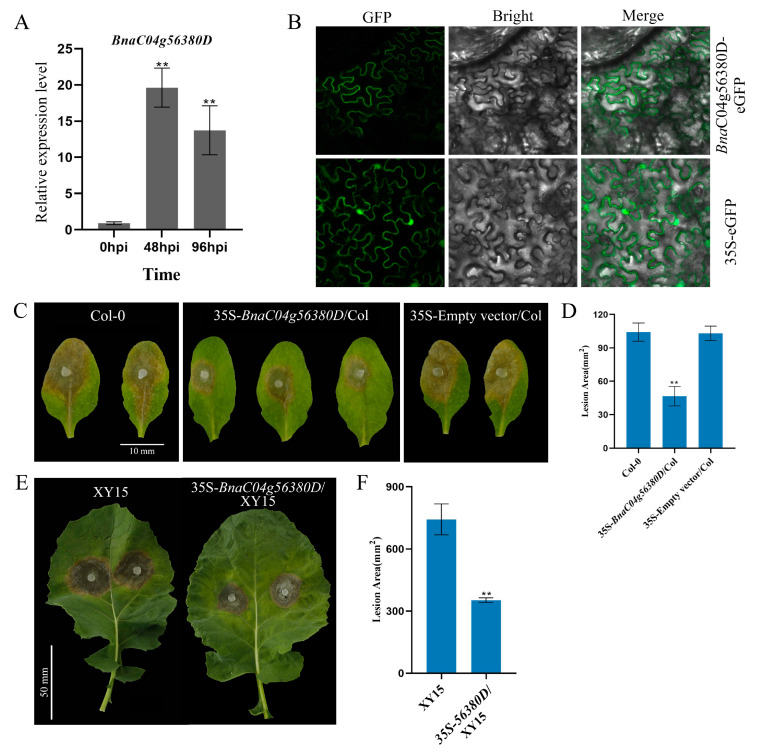
*BnaC04g56380D* overexpression enhances plant immunity to *S. sclerotiorum*. (**A**) Relative expression levels of *BnaC04g56380D* in XY15 after *S. sclerotiorum* infection for 48 or 96 h. (**B**) Subcellular localization of *Bna*C04g56380D protein. *Bna*C04g56380D-eGFP transiently expressed in *N. benthamiana* leaves with 35S-eGFP construct as negative control. Merge: Merged via GFP and bright-field microscopy. (**C**,**D**) *S. sclerotiorum* (wild-type) inoculation on detached leaves of *A. thaliana* Col-0 and 35S-*BnaC04g56380D*/Col-0. Data recorded at 48 hpi. Bar = 10 mm. ImageJ v1.51 used to analyze lesion areas. Three leaves from Col-0 or 35S-*BnaC04g56380D*/Col-0 plants selected for each infection experiment. (**E**,**F**) *S. sclerotiorum* (wild-type) inoculation on detached leaves of *B. napus* XY15 and 35S-*BnaC04g56380D*/XY15. One leaf from XY15 or 35S-*BnaC04g56380D*/XY15 selected for each infection experiment, and two mycelial plugs were inoculated on each leaf. Data recorded at 48 hpi. Bar = 50 mm. ImageJ used to analyze lesion areas. Experiments conducted three times, with similar results. Error bars represent SD (** *p* < 0.01).

**Figure 2 ijms-26-04569-f002:**
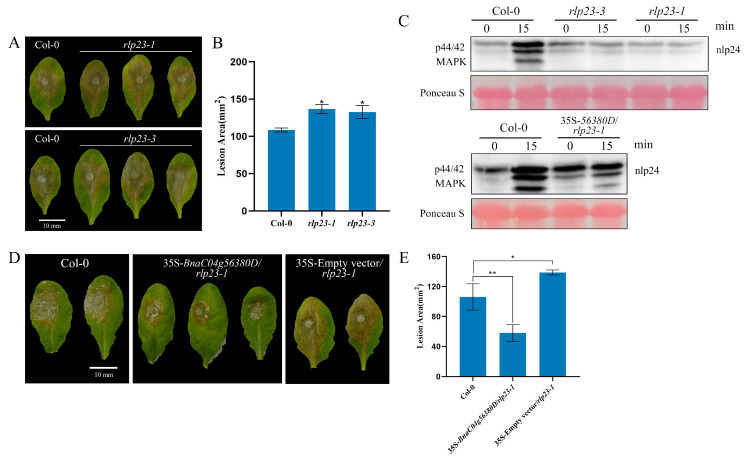
*BnaC04g56380D* functionally complements loss of *AtRLP23*. (**A**,**B**) *S. sclerotiorum* (wild type) inoculation on detached leaves of *A. thaliana* Col-0, *rlp23-1*, *rlp23-3*. Data recorded at 48 hpi. Bar = 10 mm. ImageJ used to analyze lesion areas. (**C**) nlp24-induced MAPK activation in *A. thaliana*. Seedlings treated with 1 μM of nlp24. Western blot analysis with anti-pERK antibody used for evaluation of MAPK activation. Ponceau staining served as internal reference. Experiments conducted three times, with similar results. (**D**,**E**) Lesion areas of Col-0, 35S-*BnaC04g56380D*/*rlp23-1* and 35S-Empty vector/*rlp23-1* on leaves after *S. sclerotiorum* infection. Bar = 10 mm. ImageJ used to analyze lesion areas. Three leaves of Col-0, *rlp23-1*, *rlp23-3*, or 35S-*BnaC04g56380D*/*rlp23-1* plants selected for each infection experiment. Experiments conducted three times with similar results. Error bars represent SD. Statistical significance between Col-0 and mutant analyzed using Student’s *t*-test (* *p* < 0.05, ** *p* < 0.01).

**Figure 3 ijms-26-04569-f003:**
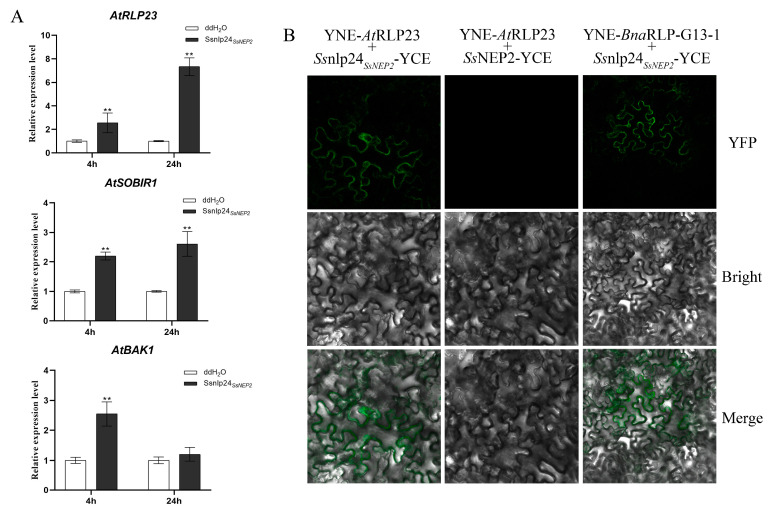
*Bna*RLP-G13-1 recognizes conserved sequences of NEP2 from *S. sclerotiorum*. (**A**) Relative expression levels of *AtRLP23*, *AtSOBIR1*, and *AtBAK1* in Col-0 after treatment with 1 μM *Ss*nlp24*_SsNEP2_* for 4 or 24 h. Experiments conducted three times, with similar results. Error bars represent SD (** *p* < 0.01). (**B**) Analysis of interactions between *At*RLP23/*Bna*RLP-G13-1 and *Ss*NEP2/*Ss*nlp24*_SsNEP2_* by BiFC in *N. benthamiana*. *At*RLP23 and *Bna*RLP-G13-1 fused to YNE, and *Ss*nlp24*_SsNEP2_* or *Ss*NEP2 fused to YCE. Merge: Merged via YFP and bright-field microscopy.

**Figure 4 ijms-26-04569-f004:**
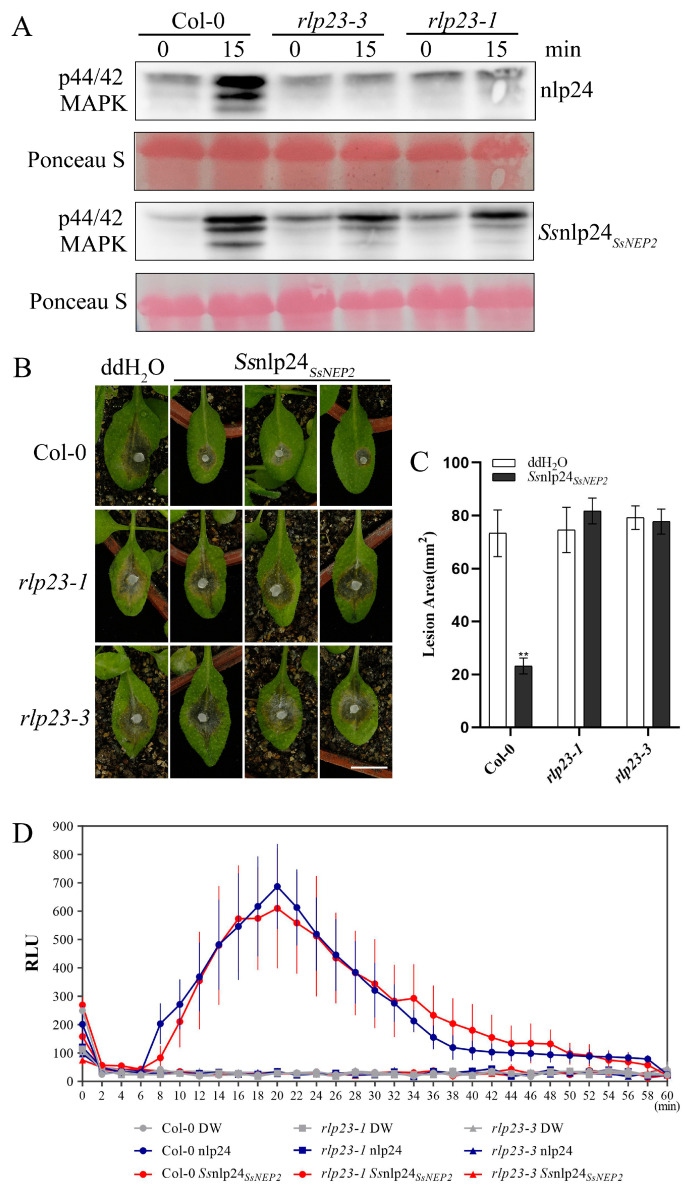
*Ss*nlp24*_SsNEP2_*induces plant resistance to *S. sclerotiorum*, which requires *At*RLP23. (**A**) The MAPK activation induced by *Ss*nlp24*_SsNEP2_* in *A. thaliana* Col-0, *rlp23-1*, and *rlp23-3*. The seedlings were treated with 1 μM nlp24 or *Ss*nlp24*_SsNEP2_*. Western blot analysis with an anti-pERK antibody was used for the evaluation of MAPK activation. Ponceau staining served as the internal reference. The experiments were conducted three times. (**B**) Four-week-old soil-grown *A. thaliana* Col-0, *rlp23-1*, and *rlp23-3* plants were pretreated with H_2_O and 1 μM *Ss*nlp24*_SsNEP2_* and inoculated with *S. sclerotiorum* (wild-type) 24 h later. The data were recorded at 36 hpi. The scale bar = 10 mm. ImageJ was used to analyze the lesion areas. (**C**) The lesion areas of *A. thaliana* Col-0, *rlp23-1*, and *rlp23-3* on the leaves after pretreatment with H_2_O and 1 μM *Ss*nlp24*_SsNEP2_* and *S. sclerotiorum* infection. ImageJ was used to analyze the lesion areas. Three leaves of the Col-0, *rlp23-1*, *rlp23-3* plants were selected for each infection experiment. The error bars represent SD (** *p* < 0.01). (**D**) Leaf disks from the true leaves of five-week-old Arabidopsis Col-0, *rlp23-1*, and *rlp23-3* mutant plants were treated with 1 µM of nlp24, 1 µM of *Ss*nlp24*_SsNEP2_*, or H_2_O. The data are reported in relative light units (RLUs) and the error bars represent SD. The experiment was repeated three times, with similar results.

**Figure 5 ijms-26-04569-f005:**
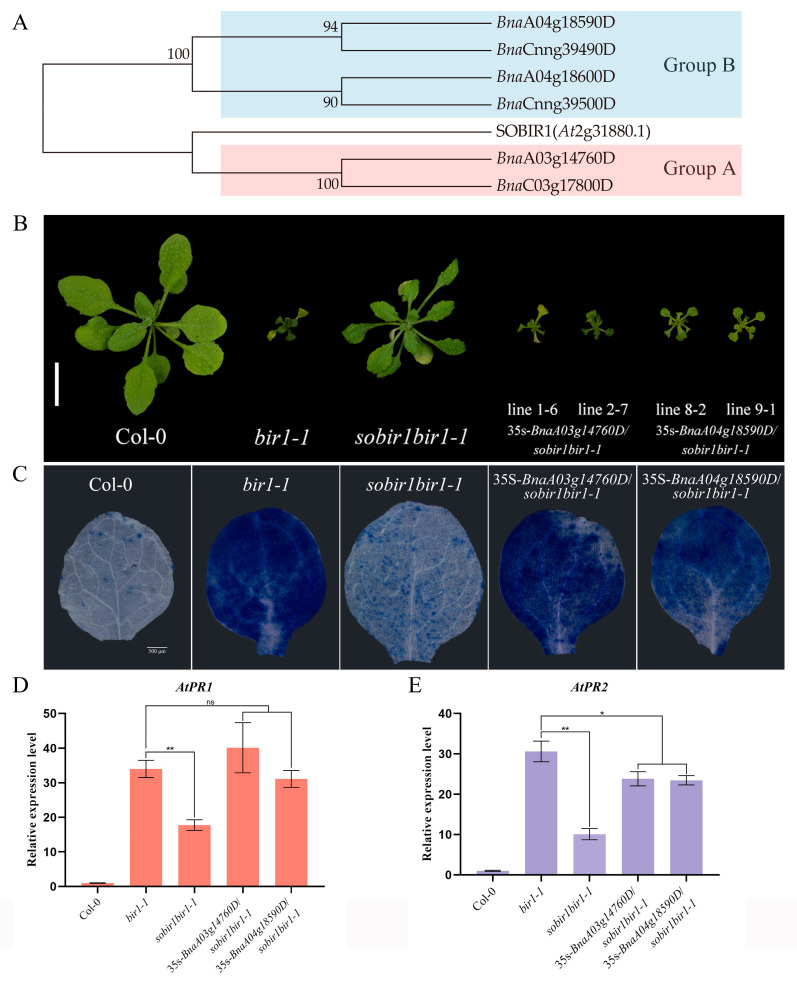
The identification of SOBIR1-encoding genes *in B. napus*. (**A**) The phylogenetic tree of SOBIR1 was constructed based on amino acid sequences from *A. thaliana* and *B. napus*. The data were downloaded from Ensembl plants (http://plants.ensembl.org/index.html, accessed on 1 January 2022). (**B**) The morphological phenotypes of the *A. thaliana* Col-0, *bir1-1*, *sobir1bir1-1*, and 35s-*BnaSOBIR1*/*sobir1bir1-1* seedlings. All the plants were grown in soil at 22 °C in parallel and photographed when they were 4 weeks old. The scale bar = 10 mm. (**C**) The DAB staining of true leaves of *A. thaliana* Col-0, *bir1-1*, *sobir1bir1-1*, and 35s-*BnaSOBIR1*/*sobir1bir1-1*. All the plants were grown on 1/2 MS medium plates at 22 °C in an incubator and stained when they were 2 weeks old. The scale bar = 500 μm. (**D**,**E**) The relative expression levels of *AtPR1* and *AtPR2* in *A. thaliana* Col-0, *bir1-1*, *sobir1bir1-1*, and 35s-*BnaSOBIR1*/*sobir1bir1-1*. The experiments were conducted three times, with similar results. The error bars represent SD (ns: non-significant, * *p* < 0.05, ** *p* < 0.01).

**Figure 6 ijms-26-04569-f006:**
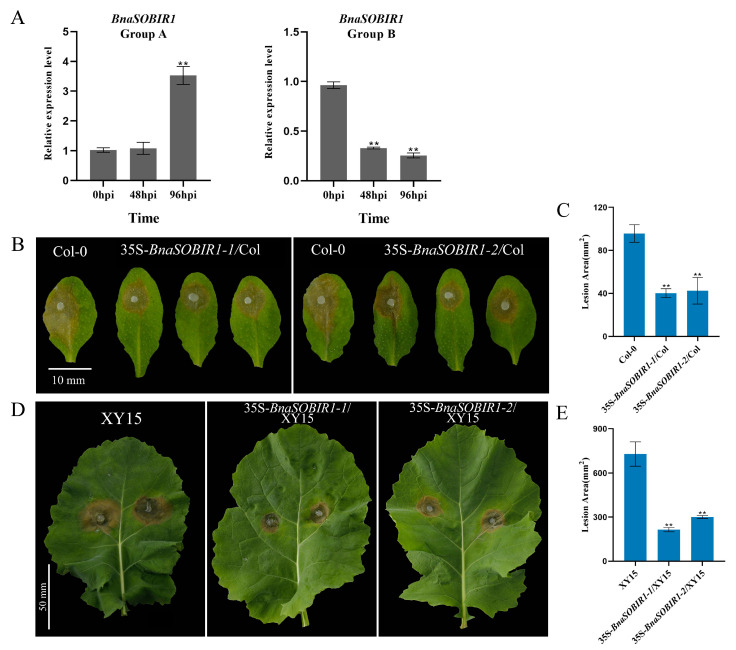
*BnaSOBIR1* enhances plant immunity to *S. sclerotiorum*. (**A**) Relative expression levels of *BnaSOBIR1* Group A and *BnaSOBIR1* Group B in XY15 after *S. sclerotiorum* infection for 48 or 96 h. (**B**,**C**) Inoculated *S. sclerotiorum* (wild-type) on detached leaves of *A. thaliana* Col-0 and 35S-*BnaSOBIR1*/Col-0. Data recorded at 48 hpi. Bar = 10 mm. ImageJ used to analyze lesioned areas. Three leaves of Col-0 or 35S-*BnaSOBIR1*/Col-0 plants selected for each infection experiment. (**D**,**E**) Inoculated *S. sclerotiorum* (wild-type) on detached *B. napus* XY15 and 35S-*BnaSOBIR1*/XY15 leaves. Data recorded at 48 hpi. One leaf from XY15 or 35S-*BnaSOBIR1*/XY15 plant selected for each infection experiment, and two mycelial plugs inoculated on each leaf. Bar = 50 mm. ImageJ used to analyze lesion areas. Experiments conducted three times, with similar results. Error bars represent SD (** *p* < 0.01).

**Figure 7 ijms-26-04569-f007:**
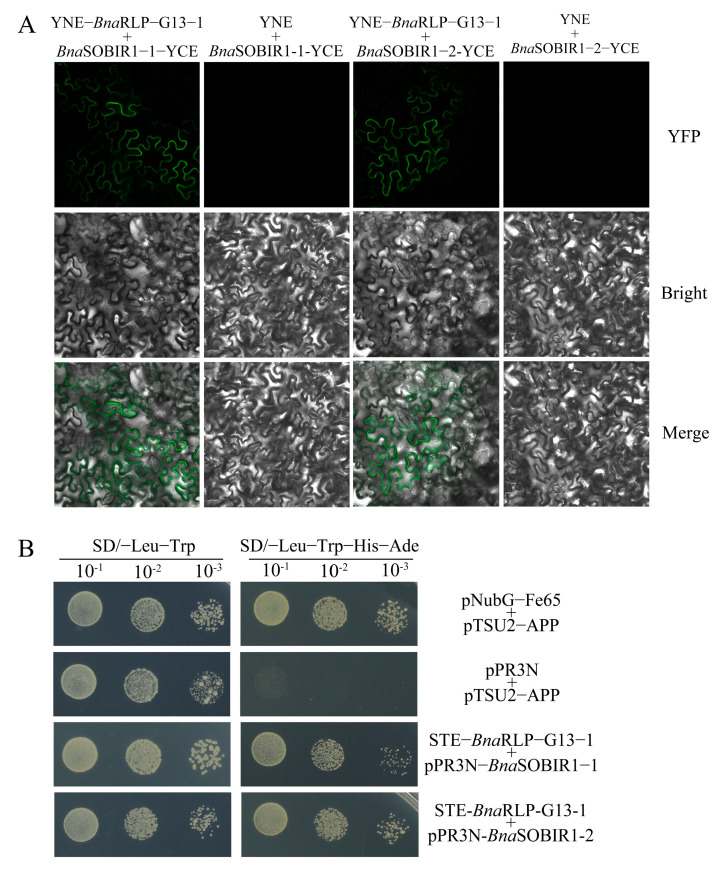
The analysis of the interactions between *Bna*RLP-G13-1 and *Bna*SOBIR1. (**A**) *Bna*RLP-G13-1 was fused to YNE, and *Bna*SOBIR1-1 and *Bna*SOBIR1-2 were fused to YCE. YNE + *Bna*SOBIR1-1-YCE and YNE + *Bna*SOBIR1-2-YCE were the negative controls. Merge: Merged via YFP and bright-field microscopy. (**B**) *Bna*RLP-G13-1 were fused to pBT3-STE, and *Bna*SOBIR1-1 and *Bna*SOBIR1-2 were fused to pPR3N. pNubG-Fe65 + pTSU2-APP served as the positive control, and pPR3N + pTSU2-APP served as the negative control. The data were recorded after 5 days.

**Figure 8 ijms-26-04569-f008:**
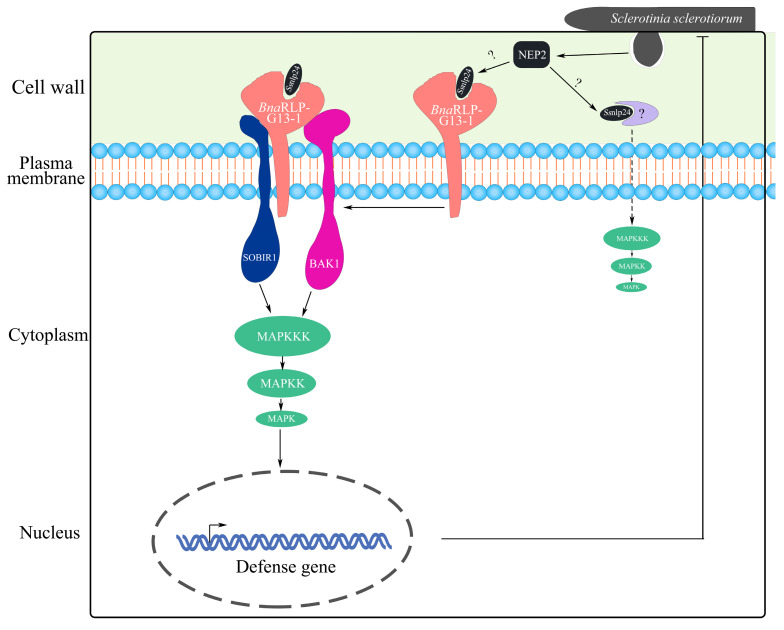
A model of *Bna*RLP-G13-1 enhancing plant resistance to *S. sclerotiorum*. *S. sclerotiorum* promotes infection by secreting a toxic protein (*Ss*NEP2). To resist the infection, plants cleave *Ss*NEP2 to generate a small peptide (*Ss*nlp24*_SsNEP2_*) through an as yet unknown mechanism. After recognizing *Ss*nlp24*_SsNEP2_*, *Bna*RLP-G13-1 forms a complex with SOBIR1 and BAK1 to activate the MAPK cascade reaction and promote immunity-related gene expression, enhancing plant resistance to *S. sclerotiorum*.

**Table 1 ijms-26-04569-t001:** Candidate genes of *BnaSOBIR1*.

Gene	Genomic DNA (bp)	Protein (aa)	Identity
*BnaA03g14760D*	1938	645	82.20%
*BnaC03g17800D*	1935	644	81.86%
*BnaCnng39490D*	2049	636	78.41%
*BnaA04g18590D*	1908	635	78.98%
*BnaA04g18600D*	1917	638	80.95%
*BnaCnng39500D*	1926	641	81.83%

## Data Availability

Data are contained within the article and Appendix A.

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
