# Peer review of "LRR Receptor-like Protein in Rapeseed Confers Resistance to Sclerotinia sclerotiorum Infection via a Conserved SsNEP2 Peptide"

_ijms, 2025, doi:10.3390/ijms26104569_

Round 1

Reviewer 1 Report (New Reviewer)

Comments and Suggestions for Authors

The manuscript, entitled: "A rapeseed LRR receptor-like protein confers resistance against Sclerotinia sclerotiorum via recognition of a conserved SsNEP2 Peptide", presents research aimed at understanding the molecular background of a very important resistance mechanism. In understanding plant-pathogen interactions, more and more information is available about PRR mechanisms. Essential components of these resistance mechanisms are the RLP and RLK molecules, which are found on the surface and in the plant cell and initiate specific immune reactions. In this case, the test plant was oilseed rape (Brassica napus), a very important pathogen of which is the fungus Sclerotinia sclerotiorum.The test plant is a Chinese variety with the designation XY15.

The introduction is sufficiently detailed and provides a good overview of the related research. The description of the materials and methods used is also adequate, both in terms of the test plants and pathogen strains, and the molecular biological methods used.

The presentation of the results is sufficiently detailed and very well illustrated. The discussion of the results and the drawing of conclusions are well-founded and can be well utilized in the resistance breeding of oilseed crops, providing additional basic knowledge.

I recommend the acceptance of the manuscript and its publication as a scientific article.

Author Response

We sincerely appreciate your thoughtful and positive evaluation of our manuscript. We are grateful for your recognition of the significance of our study and your encouraging comments regarding the clarity of our introduction, the adequacy of our methodology, and the quality of our data presentation and analysis. Thank you once again for your time and constructive feedback.

Reviewer 2 Report (New Reviewer)

Comments and Suggestions for Authors

See the attached file. This is a very good molecular research. However, it needs to be presented in an easy-to-follow and comprehensible style.

Author Response

Thank you very much for taking the time for reviewing our manuscript. We have rewritten the abstract to enhance its clarity and alignment with the manuscript's content. Additionally, we have polished the language to improve its readability and scientific tone. Please see the details in the attached file. Thank you once again for your valuable feedback. 

This manuscript is a resubmission of an earlier submission. The following is a list of the peer review reports and author responses from that submission.

Round 1

Reviewer 1 Report

Comments and Suggestions for Authors

The authors aimed to identify RLP and its co-receptor SOBIR1 from B.napus through sequence alignment followed by functional characterization in both Arabidopsis and oilseed rape. The pathogen resistance phenotypes were assessed in transgenic overexpressed plants. The interaction between RLP and SOBIR1 was further confirmed using BiFC and Y2H assays. The authors concluded that BnaRLP23 and BnaSOBIR1 form a complex to recognize the nlp peptide from S. sclerotiorum, providing insights into how B.napus perceives pathogen signals and activates downstream immune responses. Comments and suggestions are as follows:

1. The experiments depicted in Figures 1 and 2 should be more clearly and accurately described. The studies on MAPK activation, gene expression, knockout mutants, and transformants involving the BnaRLP23 were conducted in both A. thaliana and B. napus. However, the description does not always clearly specify which plant species is being used. It is important to clarify whether Arabidopsis or a member of the Brassica genus is being discussed.

2. The scale bars in Figures 2D and 6D are not clearly visible. It is recommended to adjust the images to make the scale bars more distinct.

3. For Table 1, why were only the top five genes selected for further study? Please provide a explanation for this selection.

4. In line 42, Albert et al.'s original paper used a 20-amino acid nlp20 peptide as the MAMP, whereas the authors here describe nlp24. Please provide more explanation about the differences between these peptides.

5. In line 69-76, it is suggested to revise the text as follows: "Oilseed rape is one of the main hosts of S. sclerotiorum, and the BnaRLPs family may play a crucial role in resistance to S. sclerotiorum infection. Therefore, we conducted a series of studies to analyze the regulatory role of RLP23 in the interaction between B. napus and S. sclerotiorum by searching for homologous genes of AtRLP23 in B. napus. The results indicate that BnaRLP23, in combination with BnaSOBIR1, enhances the resistance of B. napus to S. sclerotiorum by recognizing a conserved amino acid sequence derived from SsNEP2." Similar issues also appear in line 224-225.

6. The statistical analysis used to assess the significance of the lesion area results in the figures involves differential analysis. Please add details about the statistical analysis used in the Materials and Methods section, such as the type of statistical test employed to analyze data significance.

Author Response

We deeply thank you for taking the time and effort to review this manuscript. Please find our detailed responses to your comments below, with the corresponding revisions or corrections clearly highlighted or tracked in the re-submitted files.The revised files have been included as attachments.

Reviewer 2 Report

Comments and Suggestions for Authors

General comments

The MS submitted by Yang et al. "BnaRLP23 regulates Brassica napus resistance to Sclerotinia sclerotiorum by recognizing the conserved amino acid sequence from SsNEP2" aims to reveal the role of receptor-like proteins from canola in response to Sclerotinia sclerotiorum. The topic is very interesting and the manuscript contains some interesting data on the subject, however the manuscript is difficult to read due to issue with the English language. Consider working with a writer/editor that can help with language issues. Also, format lacks accuracy, I would advise a much more thorough overhaul of the manuscript before submitting it to any journal for evaluation. I began tracking individual comments on language and format issues, but it quickly became overwhelming, so I abandoned it. Below I point out a few examples.

 .- I would suggest to rewrite the abstract as in the current format is difficult to infer some of the information that the authors want to transmit. It is written in such a way that it is difficult to know what the work is about. I would recommend to rewrite it focussing on the work that has been done and summarizing it so that the reader gets a clear picture.

 .- I would suggest to change the genes nomenclature for clarifying purposes. The authors mention that four BnaRLP23 genes have been identified using AtRLP23 (T2G32680) amino acid sequence as reference for the screening, while the highest identity obtained was of 67.32%, in Arabidopsis, proteins with an identity much higher have different names. In fact, more than 50 RLP proteins have been described in Arabidopsis, thus instead of referring to four BnaRLP23 genes, and as nucleotide and the amino acid sequences are different, I would suggest to provide different names to each B. napus protein in order to avoid future nomenclature problems, specially if the similarity is low and what links them is their function, that is their interaction with a small amino acid sequence of NLPs. What they have identified are genes that share homology with AtRPL23. How many RPL23 are present in Arabidopsis? And other species? How many genes code for RPL23 in Arabidopsis? And Other species?

 .- Authors should provide information regarding the exclusion of SsNEP1 and its putative involvement in plant resistance to fungus in view of what is mentioned in Line 41. The same could be extended to peptides Ssnpl20. As mentioned by the authors “proteins could recognize the small peptide from SsNEP2”, the same could be true for SsNEP1. Also, the identification of BnaSOBIR1-1 and -2 appears in the abstract without explanation, and its implication in resistance is just a “could” and the interaction of BnaRLP23 with ssNEP2 and activation of MAPK cascades, that looks like a discussion rather than a summary for an abstract. And the last sentence of the abstract concludes after the uncertainties shown above that indeed BnaRLP23 “can” recognize, form a complex and activate to enhance plant resistance.

 .- Scientific names an gene names despite of being in the keyword section must be written in italics, and if those are proteins, the same format used in the rest of the sections should be employed, for instance “BnaSOB1R1” instead of “BnaSOB1R1”. I would advise to revise the nomenclature in the entire manuscript and use the correct format in each case.

.- How the authors explain that the lesion areas in transformants are smaller than in RPL mutants? Have those transformants been analysed for proper expression, etc? In mutants’ levels of MAPK are similar as in Col-0 at time 0, while an overexpression is shown for transformed mutants when compared with Col-0, also in Line 97, the authors are talking about infection, what experiment are they refereeing to? In Figure 1 there is no infection going on. Where are located the mutations in salk_034225 mutant, a similar mutation in the transgenes would impair the effect? Why if the lesion areas are smaller in the transformants the MAKK activation is lower than in Col-0, shouldn’t it higher at Time 15 in the transformed mutants as they manage better the response? There seems to be some discrepancy between the results shown in Figure 1D and 1F with those shown in Figure 1B. More over how the authors correlate the protection of the four transgenes with the expression levels in B. napus, as stated by them it seems that only BnaRLP23-1 is induced after infection and “may positively regulate resistance”.

.- Using constructs aimed at overexpressing certain sequences it may result in suppression of the endogenous homologous ones, have this point being evaluated?

 .- Since RLP23 is a surface receptor and SOBIR1 localizes to the plasma membrane, isn´t it normal to detect both of them in the same place as indeed is presented in Figure 7?

 I would suggest the authors to use the acronym RT-qPCR for Reverse Transcription quantitative real-time PCR.

Specific comments:

 .- Line 12: The disease caused by S. sclerotiorum in canola and other plant species is called sclerotinia stem rot (SSR) as indicated in Line 64, thus I would suggest to modify the text accordingly on that respect.

 Line14: I would suggest to write “was ectopically expressed” instead of “was ectopic expressed”. Moreover, I would specify where it was ectopically expressed.

 Line17: I would suggest to write “coding” instead of “encoding”.

 Line 18: NEP2 and MAPK are acronyms that have not been mentioned before. In line 13 RPL acronym is explained, I would suggest to do an effort in maintaining coherence with the format.

 Line 33: The acronym “PAMP” is not explained

 Line 38: The acronym “MAMP” is not explained, please check all acronyms used and explain those that have to be explained (BAK1, Eix, etc.)

 Line41: I would suggest “Pseudomonas syringae pathovar tomato (Pst) strain, DC3000, in” instead of “Pseudomonas sringae pv. tomato (P.s.t.) DC3000 in”

 Line 45: I would suggest to write “Sclerotinia sclerotiorum” instead of “S. sclerotiorum” as it is the first time that appears in the main text.

 Line 51: “conserved small peptide” and “conserved small peptide sequence” do not refer to the same thing, the authors should check carefully what they are referring to in each case.

 Line 84: What XY15 stands for? It is recommended to explain those acronyms the first time that are used, despite the fact of being mentioned in the Materials and Methods section

 Line 85: The translation of that gene was not detected by in silico analysis prior cloning? No further investigation was done to check out the correct sequence in BRAD? Why it has been included if later on it was excluded for the study but is present in Table 1 (a space is needed between Protein and (aa)) and Figure 1A?

 Line 90: Where is the mentioned secondary structure?

 Line 137: How BnaRPL23 was overexpressed in XY15? It is not clear when the authors refer to transgenic lines or agroinfiltration experiments, the nomenclature is misleading., please revised.

Line 438: Where 1 μM Ssnlp24SsNEP2 is obtained from?

Comments on the Quality of English Language

It must be improved

Author Response

We deeply thank you for taking the time and effort to review this manuscript. Please find our detailed responses to your comments below, with the corresponding revisions or corrections clearly highlighted or tracked in the re-submitted files.The responses have been included in the attachments.

Reviewer 3 Report

Comments and Suggestions for Authors

The manuscript presents novel insights, particularly regarding AtRLP32, the nlp24 small peptide, and the Arabidopsis LRR-RLK SOBIR1. In lines 67-68 the manuscript raises important scientific questions by stating, "However, the molecular mechanism of interaction between B. napus and S. sclerotiorum remains largely unknown," which is also a novel perspective. Additionally, the conclusion regarding AtRLP23 and BnaRLP23, which play crucial roles in plant immunity by recognizing a conserved amino acid sequence in SsNEP2, represents a significant discovery. 

I believe we should accept the manuscript as it is.

Author Response

We sincerely thank you for your thoughtful and encouraging feedback on our manuscript. We are delighted that you found our work on BnaRLP32, the nlp24 small peptide, and the Arabidopsis LRR-RLK SOBIR1 to offer novel insights. Your recognition of the importance of our findings, particularly regarding the molecular mechanism of interaction between B. napus and S. sclerotiorum and the roles of AtRLP23 and BnaRLP23 in plant immunity, is greatly appreciated. Your support for accepting the manuscript is highly motivating, and we are truly grateful for your positive evaluation. Thank you once again for your valuable time and consideration.

Round 2

Reviewer 2 Report

Comments and Suggestions for Authors

Despite of the efforts made by the authors, main concerns remain. See attached file

Comments on the Quality of English Language

The quality of English does limit my understanding of the research

Author Response

Thank you for your additional comments and feedback. We appreciate the time and efforts you have dedicated to reviewing our manuscript. The manuscript has already undergone professional English editing by MDPI.

We have provided detailed responses below and hopefully addressed your concerns satisfactorily. Please find the details in the attached file and let us know if further clarification is needed. Thank you again for your valuable input.

Round 3

Reviewer 2 Report

Comments and Suggestions for Authors

I do appreciate authors´s efforts to improve the manuscript, however the initial concerns remain. If the identification has been done based on sequence homology, the authors mention that the homology among RLP23 varies a lot, how they know that they have fech the right protein? If the selection is done in terms of function, why the short by sequence homology? RPL23 could be missed. If motifs are important for the search, why are not that used as filter? Too many open questions to prove with accuracy any possible hypothesis.

Comments on the Quality of English Language

Still and despite the mentioned efforts by the authors to improve the original version the manuscript needs improvement